# RETHINKING VIDEO-INRS THROUGH PERCEPTUAL OPTIMIZATION

## ABSTRACT

Implicit neural representations (INRs) have recently emerged as a powerful paradigm for video modeling, representing videos as continuous functions parameterized by network weights, rather than storing raw pixels or latent codes. Despite architectural progress, most video-INR methods largely persist with pixel-wise (MSE or $\ell_1$) losses. Through the lens of variational inference, we show—both theoretically and empirically—that these pixel-wise objectives implicitly assume Gaussian or Laplacian error distributions, which are statistically misaligned with per-video characteristics, where errors are highly structured and temporally correlated. To address this limitation, we propose shifting supervision from the pixel domain to perceptual feature spaces, which provide stable transformation spaces that relax restrictive distributional assumptions and align optimization with perceptual semantics. Specifically, we introduce two feature-domain objectives: *Multi-Vision Feature Similarity* (MVFS) for intra-frame fidelity and *Vision Subject Similarity* (VSS) for inter-frame temporal consistency. Even with a lightweight INR backbone using simple cascaded upsampling, our method surpasses state-of-the-art VAE- and diffusion-based codecs in perceptual quality while maintaining real-time decoding at an average of ~125 FPS on 1080p resolution. Our results demonstrate that perceptual supervision provides a principled and promising direction for advancing video-INRs.

## 1 INTRODUCTION

Implicit neural representations (INRs) (Sitzmann et al., 2020; Chen et al., 2021a; Su et al., 2022; Zhong et al., 2024; Kayabasi et al., 2025) have emerged as a powerful paradigm for modeling visual data as continuous functions parameterized by network weights, rather than storing discrete samples. Unlike rasterized latent representations using variational autoencoders (VAEs) (Pu et al., 2016; Ballé et al., 2018; Habibian et al., 2019), INRs directly map spatiotemporal coordinates to signal values, enabling continuous reconstruction, differentiable processing, and consistent train–test behavior. In video compression, this functional parameterization replaces raw frames with compact network weights, providing a lightweight alternative to VAEs. Methods such as NeRV (Chen et al., 2021a) and its extensions (Chen et al., 2023a; Zhao et al., 2023; Kwan et al., 2023) demonstrate that even simple feed-forward architectures can overfit any given video sequence, achieving high-quality reconstructions and supporting downstream tasks such as denoising, inpainting, and interpolation (Li et al., 2022; Zhang et al., 2024).

Despite these advantages, video-INR optimization remains dominated by pixel-level reconstruction losses. Mean squared error (MSE) is mostly used (Chen et al., 2023a; Kim et al., 2024a; Strümpler et al., 2022; Dupont et al., 2021; Zhao et al., 2023; 2024; Wu et al., 2024), with some works adopting $\ell_1$, SSIM, or MS-SSIM (Li et al., 2022; Kwan et al., 2023; Zhang et al., 2024). These choices are largely empirical, aimed at maximizing PSNR, yet rarely grounded in statistical or theoretical principles. For example, Zhao et al. (2023) observe that $\ell_1$ better preserves textures under mild motion, whereas MSE is more robust to larger motion—yet such heuristics lack theoretical justification, and the notion of "large motion" is ill-defined. Moreover, prior works have often emphasized architectural innovations (Gao et al., 2025; Tang et al., 2025; Zhu et al., 2025), sometimes at the cost of efficiency: INR models with only 1M parameters can decode slower than VAEs with over 20M parameters (Jia et al., 2025), and their compression performance still lags behind that of the most advanced VAE methods (Jiang et al., 2025; Zhang et al., 2025), limiting practical adoption.

We argue that the bottleneck lies not only in architecture but also in optimization. From a variational inference perspective, pixel-level losses correspond to log-likelihoods under fixed error distributions: MSE assumes Gaussian errors, while $\ell_1$ assumes Laplacian errors (Kingma & Welling, 2013; Goodfellow et al., 2016). While acceptable in VAEs, where dataset-level statistics are amortized, these assumptions misalign with single-video INRs, where errors are structured, temporally dependent, and video-specific. Enforcing such distributions can misguide optimization and constrain representational capacity.

This motivates the need to move beyond the raw pixel domain, and to seek transformation spaces in which error statistics are more stable and less constrained by parametric assumptions. Pretrained vision models can serve as practical proxies for these transformation spaces, as they implicitly capture semantic and structural information while relaxing rigid distributional constraints. Building on these vision models, we propose optimizing video-INRs directly in the perceptual feature domain. Specifically, we introduce two complementary feature-based objectives: *Multi-Vision Feature Similarity* (MVFS), which aggregates multiple vision backbones to enhance intra-frame fidelity, and *Vision Subject Similarity* (VSS), which enforces inter-frame temporal consistency by emphasizing subject-level representations. Even with a lightweight INR backbone using simple cascaded upsampling, our method surpasses state-of-the-art VAE- and diffusion-based codecs (Yang et al., 2022; Qi et al., 2025; Ma & Chen, 2025), while maintaining real-time decoding.

Our main contributions are summarized as follows:

- We provide a novel perspective on video-INR optimization, showing that conventional pixel-level losses implicitly assume Gaussian or Laplace error distributions, which are misaligned with single-video INRs due to strong temporal dependencies and structured content.

- We propose a feature-domain supervision framework leveraging pretrained vision models to relax restrictive distributional assumptions and align optimization with perceptual semantics. This includes *Multi-Vision Feature Similarity* (MVFS) for intra-frame fidelity and *Vision Subject Similarity* (VSS) for inter-frame consistency.

- We demonstrate that even a simple INR architecture trained with the proposed objectives achieves state-of-the-art perceptual quality, outperforming sophisticated VAE- and diffusion-based codecs, while maintaining real-time decoding at an average of $\sim$125 FPS on 1080p resolution.

## 2 RELATED WORK

**Implicit Neural Representations.** Implicit Neural Representations (INRs) (Sitzmann et al., 2020; Chen et al., 2021b; Xie et al., 2023) model signals as continuous functions that map coordinates to values, rather than storing discrete samples. This functional perspective enables resolution-agnostic reconstruction, consistent train–test behavior, and differentiable signal manipulation. In the context of compression, INRs encode raw data into compact network parameters, which can then be quantized and entropy coded (Kwan et al., 2024).

While initially popularized in 3D scene modeling (Mildenhall et al., 2021), INRs have since been adapted to diverse modalities, including images (Dupont et al., 2021; Xie et al., 2023), videos (Chen et al., 2021a; Tang et al., 2025), audio (Su et al., 2022; Lanzendörfer & Wattenhofer, 2023), and hyperspectral signals (Chen et al., 2023b; Shi et al., 2024). These extensions have enabled a wide range of applications, such as super-resolution (Zhang et al., 2022; Aiyetigbo et al., 2025), denoising (Xu & Jiao, 2023), and inpainting (Chen et al., 2023a).

**Video Implicit Neural Representations.** Video INRs extend the INR paradigm to the temporal domain by learning a function $\mathcal{F} : [0, 1]^{d_{in}} \rightarrow \mathbb{R}^{d_{out}}$ that maps temporal coordinates $t$ to frames $V_t$. In practice, this is realized via a neural decoder $\mathcal{D}$ with cascaded upsampling and nonlinearities, conditioned on temporal embeddings $\mathcal{E}$, producing reconstructed frames as $\hat{V}_t = \mathcal{D}(\mathcal{E}(t))$.

Recent architectural advances include patch-wise modeling (Maiya et al., 2023; Bai et al., 2023), temporal residual embeddings (Zhao et al., 2023), optical-flow compensation (Lee et al., 2023), hierarchical grids (Kwan et al., 2023), and adaptive backbones (Tang et al., 2025). While these strategies

enhance network capacity and reconstruction fidelity, they often introduce substantial complexity, sometimes on par with large VAE-based codecs. In contrast, the impact of optimization objectives has been relatively underexplored; we posit that reasonable loss design, alongside architectural choices, can be a critical driver of performance improvements.

**Optimization Objectives for Signal Reconstruction.** The choice of training objective fundamentally shapes neural signal reconstruction. Most INR methods rely on pixel-level losses, such as MSE or $\ell_1$, which directly penalize discrepancies in the signal space. While simple and effective for numerical fidelity, these losses implicitly assume Gaussian or Laplace error models (as analyzed in Sec. 3.1), which are systematically violated in practice, leading to heavy-tailed and video-dependent residuals. Such mismatches not only undermine the validity of optimization but also weaken the correlation with human visual perception.

A growing body of work in generative modeling highlights the limitations of pixel-level criteria: reducing distortion does not necessarily improve perceptual quality (Blau & Michaeli, 2019; Freirich et al., 2021). To address this, perceptual objectives have been proposed, typically comparing signals in learned feature spaces or leveraging adversarial discriminators. Representative approaches include LPIPS (Zhang et al., 2018), DISTS (Ding et al., 2020), and GAN-based objectives (Goodfellow et al., 2020; Darcet et al., 2023), which prioritize semantic and structural alignment over strict pixel fidelity.

Within the INR domain, perceptual optimization remains largely underexplored. Ballé et al. (2025) appear to be the first to introduce a perceptual objective for image INRs, proposing Wasserstein Distortion (WD) (Panaretos & Zemel, 2019) to improve perceptual quality compared to conventional pixel-wise training. However, the broader applicability of perceptual objectives for INRs, particularly in the video domain, remains unclear. In this work, we provide new insights into why INRs are especially amenable to feature-based supervision: even lightweight architectures, when optimized in feature spaces, can achieve state-of-the-art visual quality while maintaining real-time efficiency.

## 3 METHOD

We revisit video-INR training from a variational inference perspective, showing that pixel-level supervision is inherently misaligned with the single-video setting (Sec. 3.1). Building on this analysis, we argue that INRs are naturally suited for perceptual optimization (Sec. 3.2), and accordingly propose two feature-domain supervision strategies to enhance both intra-frame and inter-frame quality.

### 3.1 WHY PIXEL-WISE LOSSES ARE SUB-OPTIMAL FOR INRS

**Video-INRs under a Variational Framework.** Training a video-INR can be naturally cast as a rate–distortion optimization problem. The *rate* term measures model complexity (i.e., the number of bits required to encode INR parameters), while the *distortion* term quantifies reconstruction fidelity. Formally, this corresponds to approximating the true posterior $p_{\tilde{w}|x}(\tilde{w}|x)$ with a variational density $q(\tilde{w}|x)$ by minimizing the expected Kullback–Leibler (KL) divergence over the data distribution $p_x$ (Ballé et al., 2018; Kwan et al., 2024; Shi et al., 2025) (Appendix B):

$$\mathbb{E}_{x} D_{KL}[q||p_{\tilde{w}|x}] = \mathbb{E}_{x \sim p_x} \mathbb{E}_{\tilde{w} \sim q} \Big[ \underbrace{- \log p_{x|\tilde{w}}(x|\tilde{w})}_{\text{Distortion } \mathcal{L}_D} \underbrace{- \log p_{\tilde{w}}(\tilde{w})}_{\text{Rate } \mathcal{L}_R} + \log p_x(x) \Big], \quad (1)$$

where $\log p_x(x)$ is constant, thereby reducing the objective to the canonical rate–distortion trade-off.

In practice, various priors have been adopted for the rate term $p(\tilde{w})$, including uniform priors (Zhang et al., 2021b), Gaussian priors (Zhang et al., 2024; Kwan et al., 2024), and Laplacian priors (Leguay et al., 2024; Kim et al., 2024a). While the modeling of rate priors $p(\tilde{w})$ remains an open problem, our focus is on the distortion likelihood $p(x|\tilde{w})$, as it directly governs the reconstruction loss and ultimately determines the learning signal for video-INRs.

**Distributional Assumptions in Pixel-wise Losses.** Pixel-wise losses correspond to explicit assumptions on the underlying reconstruction error distribution. Minimizing an $\ell_p$ loss is equivalent

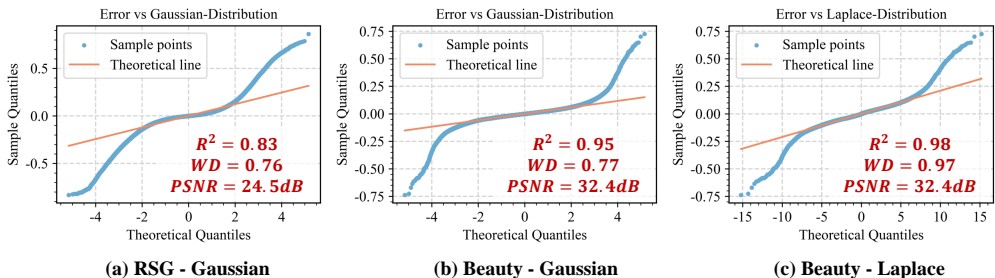

(a) RSG - Gaussian  (b) Beauty - Gaussian  (c) Beauty - Laplace

Figure 1: Q-Q (Quantile-Quantile) plots of INR reconstruction errors on UVG (Mercat et al., 2020) sequences (*ReadySteadyGo* and *Beauty*). $R^2 \uparrow$ measures the goodness of fit of the sample points to the theoretical distribution, with $R^2 = 1$ indicating perfect linear alignment, though it may underrepresent discrepancies in the tails. WD↓ (Wasserstein Distance (Panaretos & Zemel, 2019)) quantifies global distributional mismatch, and larger values indicate greater deviation. Perfect alignment would place all points on the diagonal reference line.

to performing maximum likelihood estimation (MLE) under a generalized Gaussian distribution (GGD) (Dytso et al., 2018) with shape parameter $\beta = p$:

$$p(e) = \frac{p}{2\alpha\Gamma(1/p)} \exp\left(-\left|\frac{e}{\alpha}\right|^p\right), \tag{2}$$

where $e = \boldsymbol{x} - \tilde{\boldsymbol{x}}$ denotes the reconstruction error, $\alpha$ is a scale parameter controlling the spread of the distribution, and $\Gamma(\cdot)$ is the Gamma function ensuring proper normalization. Special cases include: (a) $p = 2$, Gaussian ($\ell_2$, MSE); (b) $p = 1$, Laplacian ($\ell_1$); (c) $p < 1$, yielding heavy-tailed distributions that emphasize outliers; and (d) $p > 2$, producing sharp-peaked distributions that strongly penalize small deviations. For example, MSE minimization corresponds to Gaussian MLE:

$$\max \log p_{\boldsymbol{x}|\tilde{\boldsymbol{w}}}(\boldsymbol{x}|\tilde{\boldsymbol{w}}) = -\min \log \mathcal{N}(\boldsymbol{x}|\tilde{\boldsymbol{x}}, \sigma^2) = \min \frac{1}{2\sigma^2}\|\boldsymbol{x} - \tilde{\boldsymbol{x}}\|_2^2, \tag{3}$$

while $\ell_1$ corresponds to Laplacian MLE:

$$\max \log p_{\boldsymbol{x}|\tilde{\boldsymbol{w}}}(\boldsymbol{x}|\tilde{\boldsymbol{w}}) = -\min \log \text{Laplace}(\boldsymbol{x}|\tilde{\boldsymbol{x}}, b) = \min \frac{1}{b}\|\boldsymbol{x} - \tilde{\boldsymbol{x}}\|_1. \tag{4}$$

Thus, adopting a pixel-wise loss commits to a fixed parametric error model. In video-INRs, however, such assumptions rarely hold. Reconstruction errors are structured, temporally dependent, and highly video-specific: high-motion sequences often produce heavy tails, whereas static scenes yield more concentrated errors. Consequently, a single parametric model is inherently unreliable. By contrast, VAE-based codecs benefit from population-level amortization across datasets, making Gaussian or Laplacian assumptions more reasonable and training more stable. We provide more analysis in Appendix B.3.

**Empirical Evidence.** Fig. 1 shows Q–Q plots for *ReadySteadyGo* and *Beauty* sequences. Within a narrow range ($[-0.1, 0.1]$), empirical errors moderately align with Gaussian/Laplacian assumptions ($R^2 > 0.8$). However, extreme errors exhibit pronounced heavy tails, with Wasserstein Distance (WD) $> 0.7$, highlighting systematic mismatch.

Interestingly, the degree of mismatch roughly correlates with reconstruction quality: poorer distributional fit is associated with lower PSNR (e.g., PSNR=24.5dB with $R^2 = 0.83$ vs. PSNR=32.4dB with $R^2 = 0.95$). This supports our argument that inconsistent assumptions can misguide optimization. Previous works have likewise observed that alternative objectives (e.g., $\ell_1$, SSIM, frequency-domain losses) sometimes outperform MSE loss even under MSE evaluation (Kwan et al., 2023; Zhang et al., 2024; Kim et al., 2024b), further underscoring the pitfalls of mismatched error models. Additional analyses are provided in Appendix C.

**Remark 1.** *In summary, pixel-wise losses inherently impose rigid distributional assumptions (e.g., MSE-Gaussian) that are systematically violated in single-video INRs. Our analysis demonstrates that reconstruction errors are video-dependent, heavy-tailed, and structurally correlated, making such assumptions unreliable. These findings motivate the search for alternative representational spaces, where error statistics are more stable and better aligned with the optimization objective.*

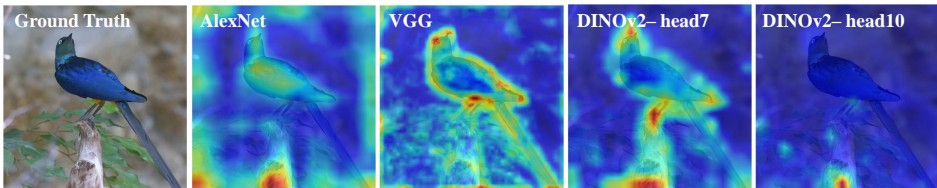

Figure 2: Illustrative heatmaps showing the feature sensitivity of different pretrained vision models on a sample sequence from YouHQ (Zhou et al., 2024). From left to right: AlexNet (Krizhevsky et al., 2012), VGG (Simonyan & Zisserman, 2014) and DINOv2 (Oquab et al., 2023)

### 3.2 TURN TO PERCEPTUAL OPTIMIZATION

**Pretrained Vision Models as Transformation.** Our preceding analysis highlights a key requirement for effective video-specific INRs: *a transformation space in which reconstruction errors exhibit more stable statistics and better align with the learning objective.* Designing such a transformation analytically is intractable due to strong temporal correlations and the lack of dataset-level averaging. As a practical solution, we propose leveraging pretrained vision models as empirical transformation functions, naturally motivating a perceptual optimization paradigm.

Unlike raw RGB supervision, perceptual objectives operate in deep feature spaces that capture semantic and structural cues—such as textures, edges, and object layouts—that correlate more closely with human perceptual sensation. Representative subjective metric examples include LPIPS (Zhang et al., 2018), which measures MSE-like distances between deep features, and DISTS (Ding et al., 2020), which integrates structural similarity with learned feature representations.

From a probabilistic perspective, feature-based objectives can be interpreted as imposing implicit distributional assumptions on reconstruction errors. For example, minimizing LPIPS is equivalent to assuming Gaussian errors in the feature space defined by the pretrained model. Crucially, these assumptions are supported by large-scale regularities captured during pretraining rather than any single video, thereby alleviating the overly restrictive distributional constraints inherent in pixel-level supervision and making perceptual optimization particularly suitable for video-specific INRs.

**Multi-Vision Feature Similarity (MVFS).** On the other hand, the pixel-to-feature mapping induced by pretrained vision models is highly nonlinear and analytically intractable. Consequently, assuming Gaussian errors in feature space does not directly translate to a corresponding distributional constraint in pixel space; such equivalence would only hold for a linear transformation. Nevertheless, even as a loose or indirect constraint, relying on a single pretrained model may bias the optimization toward specific patterns captured by that model (Fig. 2). To mitigate this potential bias and enhance generalization, we employ multiple pretrained vision models and aggregate their feature-based losses, thereby reducing the suboptimality introduced by any single inductive bias or distributional assumption. Appendix E provides a more detailed comparison of visual models.

Formally, let $\phi_m^l(\cdot)$ denote the activation of layer $l$ of the $m$-th pretrained vision model. Given reconstructed frames $\tilde{\boldsymbol{x}}$ and ground-truth frames $\boldsymbol{x}$, the MVFS loss is defined as a weighted aggregation of LPIPS- or DISTS-style distances across both models and layers:

$$\mathcal{L}_{\text{MVFS}}(\boldsymbol{x}, \tilde{\boldsymbol{x}}) = \sum_{m=1}^{M} w_m \sum_{l \in \mathcal{A}_m} d\Big(\phi_m^l(\boldsymbol{x}), \phi_m^l(\tilde{\boldsymbol{x}})\Big), \tag{5}$$

where $d(\cdot, \cdot)$ denotes the LPIPS or DISTS feature distance, $\mathcal{A}_m$ is the set of selected layers for model $m$, $w_m$ is a model-specific weight, and $M$ is the total number of vision models.

**Vision Subject Similarity (VSS).** Besides, a common challenge in video compression is the neglect of inter-frame stationarity, which can produce noticeable quality fluctuations across frames (Sec. 4.1). This issue is exacerbated under perceptual optimization. Inspired by temporal consistency assessment techniques (Huang et al., 2024), we aim to enforce subject-level consistency across frames. To this end, we leverage DINOv2 (Oquab et al., 2023) to extract features that capture subject

**(a) Encoding (Training)**

embedding

encoder → 1 2 3 → decoder

Vision Models
(VGG, Alexnet, DINOv2,...)

MVFS and VSS Loss

😊 Mitigate inductive bias

Transmission

**(b) Decoding**

1 2 3

decoder

Forward

😊 Perceptual quality

GOOD

Figure 3: Overview of the proposed video-INR framework. The encoding is equivalent to training the network, where the video signal is parameterized as a neural function. During decoding, only a forward pass through the trained decoder is required for reconstruction.

identity. Unlike standard classification networks, which are trained to collapse intra-class variations, DINOv2 is self-supervised and produces representations that are both semantically meaningful and sensitive to subtle identity differences (Vanyan et al., 2023). This property makes DINOv2 features particularly suitable for measuring identity consistency across frames.

Formally, we extract its DINOv2 feature $f_t$ and define the subject consistency loss as:

$$\mathcal{L}_{\text{VSS}} = \sum_{i \in \mathring{U}(t,\delta)} \text{dist}(f_i, f_t), \tag{6}$$

where $\mathring{U}(t,\delta)$ denotes the temporal neighborhood of frame $t$ (excluding $t$ itself), and $\text{dist}(\cdot, \cdot)$ is a feature distance metric such as cosine similarity.

**Overall Pipeline.** Our framework is illustrated in Fig. 3. In the encoding stage, each video frame $V_t$ is first mapped into a compact, high-dimensional embedding via an encoder. The embeddings are then decoded by a lightweight feed-forward decoder—composed of a few cascaded upsampling layers (Fig. 4)—to reconstruct the frame $\tilde{V}_t$. For transmission, only the embeddings and decoder weights are quantized and entropy-coded. At the receiver side, a forward pass through the decoder suffices to reconstruct the video. Detailed network specifications are provided in Appendix D.

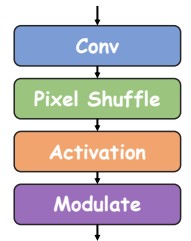

For training, we jointly optimize over multiple feature spaces (e.g., AlexNet, VGG, DINOv2, GAN) using the proposed MVFS and VSS supervision:

$$\mathcal{L}(\phi) = \lambda_1 \mathcal{L}_{\ell_1} + \lambda_2 \mathcal{L}_{\text{SSIM}} + \lambda_3 \mathcal{L}_{\text{MVFS}} + \lambda_4 \mathcal{L}_{\text{VSS}} + \lambda_5 \mathcal{L}_{\text{GAN}}, \tag{7}$$

Figure 4: Upsampling layer.

where, following successful practices in image and video reconstruction (Snell et al., 2017; Zhang et al., 2024; Yao et al., 2025), $\ell_1$ and SSIM (Wang et al., 2004) losses are included as regularization terms to stabilize training and preserve low-level fidelity.

Remarkably, despite the simplicity of the architecture—without sophisticated temporal prediction, patch-wise modeling, or hierarchical grid structures—our method achieves performance surpassing state-of-the-art VAE- and diffusion-based approaches. This demonstrates the effectiveness of *perceptually grounded optimization* for advancing video-INRs.

## 4 EXPERIMENTS

We compare with representative perceptual-optimized methods from three paradigms: (1) *GAN-based*: PLVC (Yang et al., 2022), which employs adversarial learning to enhance perceptual quality; (2) *VAE-based*: GLC (Qi et al., 2025), which introduces perceptual coding in the generative latent space and achieves state-of-the-art perceptual performance, and DVC-P (Zhang et al., 2021a), a early representative approach exploring perceptual representation; (3) *Diffusion-based*: DiffVC (Ma & Chen, 2025), a state-of-the-art approach that leverages inter-frame diffusion priors for improved perceptual reconstruction.

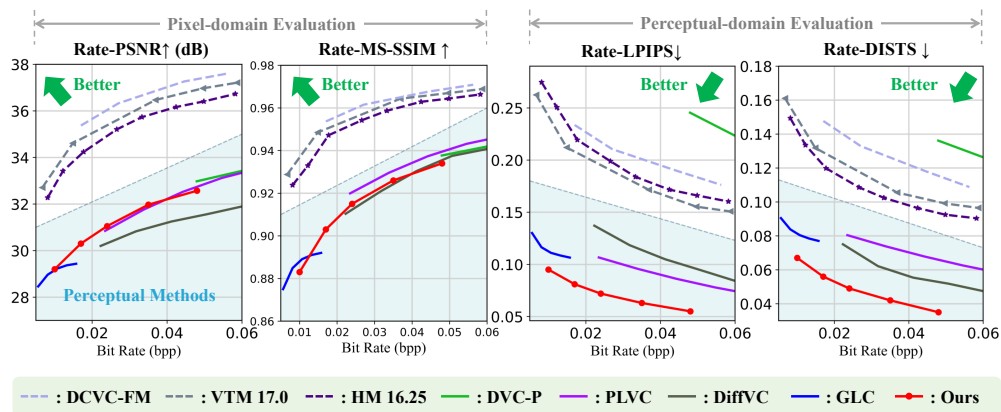

Figure 5: Rate–distortion curves on the UVG dataset. Solid lines denote perceptual-optimized methods; dashed lines denote pixel-optimized codecs. Our method achieves the best perceptual quality.

Table 1: Frame-level BD-metrics (Bjontegaard, 2001) on the UVG dataset with HM as the anchor. Decoding efficiency is reported in FPS; for our method, both "FP16 (FP32)" are provided, while other learning-based methods are in FP16. Symbols ↑ / ↓ indicate that higher/lower values are better. The best results under perceptual optimization are highlighted in **bold**, and our method is additionally indicated with a gray background.

| Methods | Type | PSNR↑ | MS-SSIM↑ | LPIPS↓ | DISTS↓ | Dec. FPS↑ |
|---|---|---|---|---|---|---|
| **HandCraft** | | | | | | |
| HM 16.25 (Anchor) (HM) | H.265 | 0. | 0. | 0. | 0. | ∼40 |
| VTM 17.0 (VTM) | H.266 | 0.68 | 0.006 | -0.017 | -0.056 | ∼25 |
| **Pixel Optimization** | | | | | | |
| DHVC 2.0 (Lu et al., 2024) | HVAE | 0.50 | 0.005 | / | / | <8 |
| DCVC-FM (Li et al., 2024) | VAE | 1.06 | 0.006 | 0.015 | 0.024 | <5 |
| NVRC (Kwan et al., 2024) | INR | 1.25 | 0.010 | / | / | <25 |
| **Perceptual Optimization** | | | | | | |
| PLVC (Yang et al., 2022) | GAN | -3.80 | **-0.026** | -0.088 | -0.029 | / |
| GLC (Qi et al., 2025) | VAE | -3.92 | -0.043 | -0.123 | -0.056 | < 5 |
| DVC-P (Zhang et al., 2021a) | VAE | **-3.37** | -0.026 | 0.071 | 0.040 | / |
| DiffVC (Ma & Chen, 2025) | Diffusion | -4.80 | -0.034 | -0.069 | -0.041 | <0.1 |
| Ours | INR | -3.8$_{\pm0.3}$ | -0.04$_{\pm0.01}$ | **-0.13**$_{\pm0.01}$ | **-0.064**$_{\pm0.01}$ | **125(75)**$_{\pm5}$ |

In addition, we evaluate strong pixel-optimized baselines such as DCVC (Li et al., 2021) DHVC 2.0 (Lu et al., 2024), DCVC-FM (Li et al., 2024), HNeRV (Chen et al., 2023a), and NVRC (Kwan et al., 2024), along with conventional video codecs H.265/HEVC (Sullivan et al., 2012) and H.266/VVC (Bross et al., 2021) for reference. More details and results are provided in Appendix F.

## 4.1 MAIN RESULTS

**Frame-level Evaluation.** We adopt four widely used metrics for evaluation: PSNR and MS-SSIM to measure pixel-level fidelity, and LPIPS and DISTS to assess perceptual similarity. As illustrated in Fig. 5, our method consistently improves perceptual metrics (LPIPS and DISTS) while maintaining competitive performance in the pixel domain compared to perceptually optimized baselines. Table 1 reports the corresponding BD-metrics (Bjontegaard, 2001), summarizing the average performance differences across bitrates. Minor interpolation errors may occur because prior works sometimes report results at slightly different bitrates, but overall trends remain consistent. Notably, our approach achieves state-of-the-art perceptual quality while offering a substantial advantage in decoding speed, reaching on average 75 FPS in FP32 and 125 FPS in FP16 across UVG sequences under serial decoding, which is sufficient for real-time playback scenarios.

A key advantage of INR-based methods is their inherent decoding parallelism. Existing VAE-based approaches rely on inter-frame references, which limit parallel decoding and increase latency. In contrast, our method is fully parallelizable: in principle, all frames can be decoded in a single forward pass, with total latency equivalent to a single model evaluation. More complexity analysis is provided in the Appendix F.3.

While video-INR methods are sometimes criticized for high encoding latency that constrains live streaming (Bentaleb et al., 2025), our perceptually optimized INR remains highly suitable for video-on-demand (VOD) (Liu et al., 2024) and large-scale storage applications, where decoding efficiency constitutes the primary bottleneck.

Table 2: Sequence-level evaluation on UVG using VBench (Huang et al., 2024).

| Methods | Type | Bpp↓ | Subject Consistency↑ | Background Consistency↑ | Motion Smoothness↑ | Average Score↑ |
|---|---|---|---|---|---|---|
| DCVC-FM (Li et al., 2024) | VAE | 0.051 | 84.5% | **90.6%** | **99.2%** | 91.4% |
| DCVC (Li et al., 2021) | VAE | 0.025 | **85.5%** | 90.0% | 99.1% | **91.5%** |
| HNeRV (Chen et al., 2023a) | INR | 0.010 | 80.1% | 88.3% | 98.5% | 89.0% |
| Ours | INR | 0.010 | $84.5_{\pm1.5}$% | $89.4_{\pm1.0}$% | $99.0_{\pm0.5}$% | $91.0_{\pm1.8}$% |
| Empirical Min | / | / | 14.62% | 26.15% | 70.60% | / |
| Empirical Max | / | / | 100% | 100% | 99.75% | / |

**Sequence-level Evaluation.** Frame-level metrics, though widely used, fail to capture temporal dynamics across video sequences. VAE-based methods often exhibit fluctuations in frame quality due to inter-frame dependencies, which remain hidden under conventional evaluations. This limitation is particularly pronounced under perceptual optimization, where human viewers are highly sensitive to temporal inconsistencies.

Prior studies (Huang et al., 2024; Zheng et al., 2025) have shown that commonly used video-level metrics—such as Inception Score (IS) (Salimans et al., 2016), Fréchet Video Distance (FVD) (Unterthiner et al., 2018), and CLIPSIM (Radford et al., 2021)—often do not align well with subjective judgments. Here, we adopt the recently proposed VBench (Huang et al., 2024), which provides a more reliable, fine-grained sequence-level evaluation of perceptual video quality.

Using VBench, we benchmark our method alongside popular VAE- and INR-based approaches on UVG (Table 2). Our method achieves comparable temporal consistency while maintaining a lower bitrate. We encourage future work to combine sequence-level and frame-level metrics for a more comprehensive assessment. Detailed definitions and additional results are provided in Appendix F.2.

Table 3: Loss ablation on subset from YouHQ (Zhou et al., 2024).

| | Pixel Optimization | | | Perceptual Optimization | | |
|---|---|---|---|---|---|---|
| | MSE | $\ell_1$ | $\ell_1$+SSIM | w/ VFS (VGG) | w/ MVFS | w/ MVFS & VSS |
| PSNR | 36.21 | 35.34 | **36.51** | 34.41 | 34.61 | 34.75 |
| MS-SSIM | 0.9742 | 0.9773 | **0.9824** | 0.9756 | 0.9774 | 0.9782 |
| LPIPS | 0.1630 | 0.1702 | 0.1496 | 0.0284 | 0.0225 | **0.0190** |
| DISTS | 0.1362 | 0.1414 | 0.1235 | 0.0152 | 0.0103 | **0.0085** |

## 4.2 ABLATION STUDY

**Loss Ablation.** Table 3 reports results on a subset of YouHQ (Zhou et al., 2024). Among pixel-level objectives, combining $\ell_1$ with SSIM achieves the highest PSNR and MS-SSIM, showing that structural regularization helps preserve low-level details; however, LPIPS and DISTS remain relatively high, indicating limited perceptual fidelity. Introducing perceptual optimization substantially reduces these perceptual distances. A single Vision Feature Similarity (VFS) using VGG, similar to LPIPS, already improves similarity in feature space, while MVFS aggregation across multiple models further alleviates the bias of any single feature extractor. Incorporating VSS enforces temporal consistency, leading to the best overall results, with LPIPS reduced to 0.019 and DISTS to 0.0085.

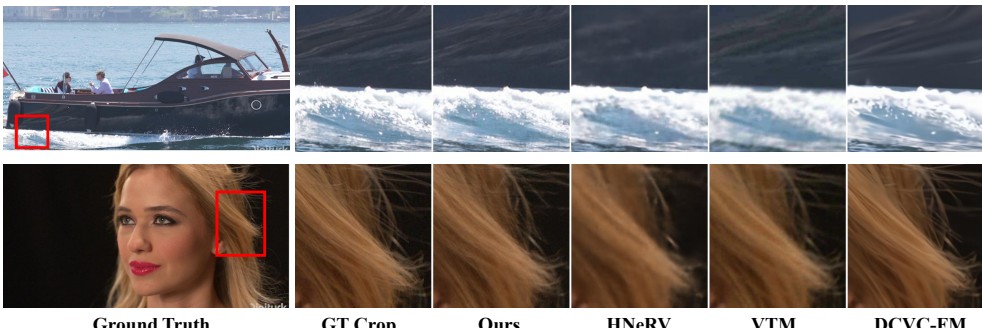

Figure 6: Visual comparison on a UVG sequence at $0.014$ bpp ($\sim 1700\times$ compression).

**Visualization.** Qualitative comparisons with HNeRV (INR), DCVC-FM (VAE), and VVC (traditional codec) are shown in Fig. 6. Our method produces sharper reconstructions and preserves richer textures even under extreme compression, whereas competing approaches often suffer from oversmoothing or perceptual artifacts. Additional visualizations are provided in the Appendix F.4.

## 5 LIMITATIONS AND FUTURE WORK

To maintain fast decoding, we adopt a lightweight feed-forward architecture without advanced temporal or hierarchical modules. While this simplicity enables real-time inference, it inevitably limits the achievable rate–distortion tradeoff. Future work could explore more advanced architectural components, such as hierarchical grids (Kwan et al., 2023), context modeling (Zhang et al., 2024; Kwan et al., 2024), or learned quantization strategies (Shi et al., 2025), which may further improve efficiency and scalability.

Another limitation concerns the cost of perceptual optimization. Our framework relies on multiple pretrained vision models to transform supervision from the pixel domain to feature spaces. Although this mitigates model-specific biases and improves robustness, it substantially increases training complexity due to gradient computation. The overhead depends on the chosen models, but under our current setting, training time grows by more than $2\times$ comapred with naive pixel-wise supervision. Developing lightweight yet reliable perceptual surrogates, therefore, remains an important direction.

Beyond compression, our framework suggests broader applications of perceptually grounded optimization. In particular, integrating INR with feature-space supervision could benefit real-world video understanding tasks such as interpolation and classification, where maintaining semantic and temporal consistency is equally critical.

## 6 CONCLUSION

We revisited the optimization of video-INRs through the lens of variational inference and highlighted a key limitation of conventional pixel-based losses: their reliance on simplistic error models that are statistically mismatched with single-video reconstruction. Our central insight is to leverage multiple pretrained vision models as proxy transformations, shifting supervision from the pixel domain to the feature domain and thereby alleviating inductive biases. Building on this idea, we introduced a perceptual feature-domain scheme comprising Multi-Vision Feature Similarity (MVFS) for intra-frame fidelity and Vision Subject Similarity (VSS) for inter-frame consistency. This formulation relaxes the restrictive assumptions of pixel-domain objectives and aligns INR optimization with perceptual semantics. Extensive experiments show that even with a lightweight INR backbone based on cascaded upsampling layers, our method substantially improves perceptual quality over both state-of-the-art VAE- and diffusion-based baselines, and achieves real-time decoding, averaging $\sim 125$ FPS at 1080p resolution. These findings suggest that perceptual supervision offers a promising new direction for video-INRs: shifting emphasis from architectural sophistication to optimization principles, and paving the way for broader applications of INRs beyond compression.

## ETHICS STATEMENT

This work investigates perceptual video compression with the goal of improving storage and transmission efficiency while preserving visual quality. All experiments are conducted on publicly available datasets, ensuring that no private or sensitive data is used. We acknowledge that compression inevitably introduces distortions, which may affect interpretation in safety-critical or high-stakes applications. We encourage responsible use of the proposed methods and careful consideration of potential risks when deploying them in such domains.

## REPRODUCIBILITY STATEMENT

We provide detailed descriptions of model architectures and evaluation settings in the Appendix. All experiments were conducted under a consistent environment with fixed random seeds to ensure reproducibility. To further facilitate replication and extension, we will publicly release the complete codebase upon publication.

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

## A The Use of LLM

We used large language models (LLMs) solely for language refinement, including grammar checking and phrasing improvement. All scientific contributions—including methods, experiments, code, and conceptual design—were independently developed by the authors. LLMs influenced only textual clarity, not technical content.

## B INR in Variational Viewpoint

Variational inference has long served as a theoretical foundation for probabilistic representation learning (Kingma & Welling, 2013; Ballé et al., 2018). In this subsection, we reinterpret implicit neural representations (INRs) from a variational perspective, and show why the conventional practice of training INRs with pixel-wise losses is sub-optimal.

### B.1 From VAE to INR

In variational autoencoders (VAEs), one constructs a stochastic mapping between the latent variable $z$ and the signal $x$, such that $x \sim p_\theta(x|z)$, while inference corresponds to approximating the posterior $p_\theta(z|x)$. In contrast, INRs directly parameterize the signal $x$ as a function of weights $w$ and coordinates $t$, i.e.,

$$\boldsymbol{x} = \mathcal{F}(\boldsymbol{w}, \boldsymbol{t}), \tag{8}$$

where $\boldsymbol{t}$ denotes spatial or temporal coordinates. Thus, while VAEs adopt a *latent-centric* representation, INRs implement a *function-centric* representation: the role of the latent variable $z$ is now played by the neural weights $w$ that uniquely encode the signal. This connection allows us to view INR training as a special case of variational inference over function such that $x \sim p_{(x|w)}$, while inference corresponds to approximating the posterior $p_{(w|x)}$. $w$ is largely determined by optimization-induced distribution constrain and inherent architecture-induced inhertic bias.

### B.2 Variational formulation

Formally, this corresponds to approximating the true posterior $p_{\tilde{\boldsymbol{w}}|\boldsymbol{x}}(\tilde{\boldsymbol{w}}|\boldsymbol{x})$ with a variational density $q(\tilde{\boldsymbol{w}}|\boldsymbol{x})$ by minimizing the expected Kullback–Leibler (KL) divergence over the data distribution $p_{\boldsymbol{x}}$ (Ballé et al., 2018; Kwan et al., 2024; Shi et al., 2025):

$$\mathbb{E}_{\boldsymbol{x} \sim p_{\boldsymbol{x}}} D_{KL}[q||p_{\tilde{\boldsymbol{w}}|\boldsymbol{x}}] = \mathbb{E}_{\boldsymbol{x} \sim p_{\boldsymbol{x}}} \mathbb{E}_{\tilde{\boldsymbol{w}} \sim q} \left[ \log \frac{q(\tilde{\boldsymbol{w}}|\boldsymbol{x})}{p_{\tilde{\boldsymbol{w}}|\boldsymbol{x}}(\tilde{\boldsymbol{w}}|\boldsymbol{x})} \right] \tag{9}$$

$$= \mathbb{E}_{\boldsymbol{x} \sim p_{\boldsymbol{x}}} \mathbb{E}_{\tilde{\boldsymbol{w}} \sim q} \left[ \log q(\tilde{\boldsymbol{w}}|\boldsymbol{x}) - \log p_{\tilde{\boldsymbol{w}}|\boldsymbol{x}}(\tilde{\boldsymbol{w}}|\boldsymbol{x}) \right]. \tag{10}$$

Applying Bayes' theorem, the posterior can be written as

$$p_{\tilde{\boldsymbol{w}}|\boldsymbol{x}}(\tilde{\boldsymbol{w}}|\boldsymbol{x}) = \frac{p_{\boldsymbol{x}|\tilde{\boldsymbol{w}}}(\boldsymbol{x}|\tilde{\boldsymbol{w}}) p_{\tilde{\boldsymbol{w}}}(\tilde{\boldsymbol{w}})}{p_{\boldsymbol{x}}(\boldsymbol{x})}. \tag{11}$$

Substituting into Eq. 10 yields:

$$\mathbb{E}_{\boldsymbol{x} \sim p_{\boldsymbol{x}}} D_{KL}[q||p_{\tilde{\boldsymbol{w}}|\boldsymbol{x}}] = \mathbb{E}_{\boldsymbol{x} \sim p_{\boldsymbol{x}}} \mathbb{E}_{\tilde{\boldsymbol{w}} \sim q} \big[ \log q(\tilde{\boldsymbol{w}}|\boldsymbol{x}) - \log p_{\boldsymbol{x}|\tilde{\boldsymbol{w}}}(\boldsymbol{x}|\tilde{\boldsymbol{w}}) - \log p_{\tilde{\boldsymbol{w}}}(\tilde{\boldsymbol{w}}) + \log p_{\boldsymbol{x}}(\boldsymbol{x}) \big]. \tag{12}$$

We now examine each component in turn.

1. $\mathbb{E}[\log q(\tilde{\boldsymbol{w}}|\boldsymbol{x})]$. In practice, inference in coding can be related with quantization. By treating quantization as adding uniform noise (Ballé et al., 2017), we have

$$q(\tilde{\boldsymbol{w}}|\boldsymbol{x}) = \prod_i \mathcal{U}(\tilde{\boldsymbol{w}}_i | \boldsymbol{w}_i - \tfrac{1}{2}, \boldsymbol{w}_i + \tfrac{1}{2}), \quad \boldsymbol{w} = \mathcal{F}^{-1}(\boldsymbol{x}), \tag{13}$$

where $\mathcal{U}$ denotes a uniform distribution. Its expectation is constant, and can be safely ignored.

2. $-\mathbb{E}[\log p_{\boldsymbol{x}|\tilde{\boldsymbol{w}}}(\boldsymbol{x}|\tilde{\boldsymbol{w}})]$. This term enforces distributional alignment between the reconstruction induced by $\tilde{\boldsymbol{w}}$ and the original signal $\boldsymbol{x}$. In coding terminology, it directly corresponds to the *distortion* term: the better $\tilde{\boldsymbol{w}}$ explains $\boldsymbol{x}$, the lower the penalty. Importantly, if the likelihood model $p_{\boldsymbol{x}|\tilde{\boldsymbol{w}}}$ is chosen as a Gaussian with fixed variance, minimizing this term reduces to minimizing mean squared error (MSE). This explains why pixel-wise losses emerge naturally but also highlights their limitations: they implicitly assume Gaussian residuals, which are mismatched with the structured errors in INR-based reconstructions.

3. $-\mathbb{E}[\log p_{\tilde{\boldsymbol{w}}}(\tilde{\boldsymbol{w}})]$. This term reflects the complexity of representing the signal in the function space, i.e., the *rate* cost in coding. A more compact or structured prior on $\tilde{\boldsymbol{w}}$ leads to improved efficiency, and directly relates to the regularization of INR weights or their quantization in compression frameworks.

4. $\mathbb{E}[\log p_{\boldsymbol{x}}(\boldsymbol{x})]$. This term is constant for a given sequence and can be discarded from the optimization objective.

Collecting the non-trivial terms, we obtain:

$$\mathcal{L} = \mathcal{L}_R + \lambda \mathcal{L}_D = -\log p_{\tilde{\boldsymbol{w}}}(\tilde{\boldsymbol{w}}) - \log p_{\boldsymbol{x}|\tilde{\boldsymbol{w}}}(\boldsymbol{x}|\tilde{\boldsymbol{w}}), \tag{14}$$

where $\mathcal{L}_R$ captures the rate (complexity of the representation) and $\mathcal{L}_D$ denotes the distortion (distributional alignment between reconstructed and original signals).

### B.3 DISTRIBUTIONAL ASSUMPTIONS IN PIXEL-WISE LOSSES.

Pixel-wise losses correspond to explicit distributional assumptions on the reconstruction error (Murphy, 2012; Kingma & Welling, 2013; Goodfellow et al., 2016). More precisely, minimizing an $\ell_p$ loss is equivalent to maximum likelihood estimation (MLE) under a generalized Gaussian distribution (GGD) (Dytso et al., 2018), whose probability density function is given by:

$$p(e) = \frac{p}{2\alpha\Gamma(1/p)} \exp\left(-\left|\frac{e}{\alpha}\right|^p\right), \tag{15}$$

where $e = \boldsymbol{x} - \tilde{\boldsymbol{x}}$ denotes the reconstruction error, $\alpha > 0$ is the scale parameter controlling dispersion, $p = \beta$ is the shape parameter determining tail heaviness and peak sharpness, and $\Gamma(\cdot)$ denotes the Gamma function, defined as

$$\Gamma(z) = \int_0^\infty t^{z-1} e^{-t} dt, \quad z > 0. \tag{16}$$

Special cases of the GGD include: (a) $p = 2$: Gaussian distribution $\mathcal{N}(0, \sigma^2)$, leading to mean squared error (MSE); (b) $p = 1$: Laplace distribution $\mathrm{Laplace}(0, b)$, leading to $\ell_1$ loss; (c) $p < 1$: heavy-tailed distributions with higher robustness to outliers; (d) $p > 2$: sharper-peaked distributions emphasizing small errors. For instance, Gaussian error modeling leads to:

$$\max \log p_{\boldsymbol{x}|\tilde{\boldsymbol{w}}}(\boldsymbol{x}|\tilde{\boldsymbol{w}}) = -\min \log \mathcal{N}(\boldsymbol{x}|\tilde{\boldsymbol{x}}, \sigma^2) \tag{17}$$

$$= -\min \log \frac{1}{\sigma\sqrt{2\pi}} \exp\left(-\frac{1}{2\sigma^2}||\boldsymbol{x} - \tilde{\boldsymbol{x}}||^2\right) \tag{18}$$

$$= \min \frac{1}{2\sigma^2}||\boldsymbol{x} - \tilde{\boldsymbol{x}}||^2, \tag{19}$$

while Laplace error modeling corresponds to:

$$\max \log p_{\boldsymbol{x}|\tilde{\boldsymbol{w}}}(\boldsymbol{x}|\tilde{\boldsymbol{w}}) = -\min \log \mathrm{Laplace}(\boldsymbol{x}|\tilde{\boldsymbol{x}}, b) \tag{20}$$

$$= -\min \log \frac{1}{2b} \exp\left(-\frac{1}{b}||\boldsymbol{x} - \tilde{\boldsymbol{x}}||_1\right) \tag{21}$$

$$= \min \frac{1}{b}||\boldsymbol{x} - \tilde{\boldsymbol{x}}||_1. \tag{22}$$

Thus, adopting a pixel-wise loss is equivalent to committing to a fixed parametric error model. However, in video-INRs such assumptions rarely hold. First, reconstruction errors deviate significantly from Gaussian or Laplacian distributions due to strong temporal dependencies and structured spatial patterns. Second, error statistics are highly video-dependent: high-motion videos often produce

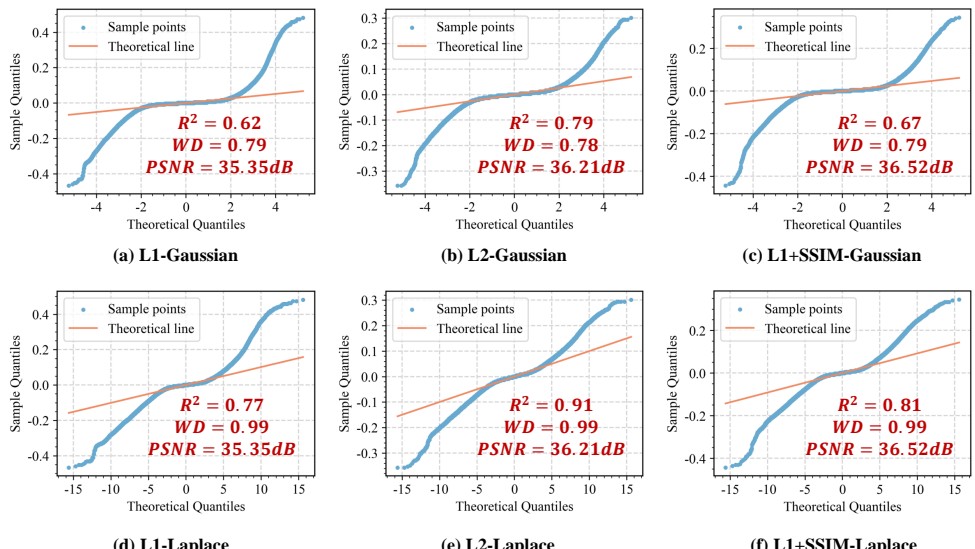

Figure 7: Q–Q (Quantile–Quantile) plots of reconstruction errors trained with different pixel-level loss functions on a sample (Fig. 2) from YouHQ (Zhou et al., 2024). Each plot compares empirical error distributions with their theoretical counterparts (Gaussian or Laplace). $R^2 \uparrow$ measures the linear alignment with the reference distribution ($R^2 = 1$ indicates perfect fit, though tail deviations may be underestimated). $WD \downarrow$ denotes the Wasserstein Distance (Panaretos & Zemel, 2019), where larger values indicate stronger distributional mismatch. Perfect distributional alignment would result in points lying along the diagonal reference line.

heavier-tailed residuals, while static scenes yield more concentrated error profiles. Consequently, a single parametric assumption (e.g., Gaussian residuals) is inherently unreliable and leads to suboptimal optimization objectives. This motivates the need for alternative formulations that relax fixed distributional assumptions and better capture the true statistics of INR reconstruction errors.

**Remark 2.** *It is important to emphasize that once a perceptual transform is introduced, or when the distortion measure deviates from a Euclidean form, the direct correspondence between minimizing a distortion metric and performing maximum likelihood estimation no longer strictly holds.*

## C  ERROR DISTRIBUTION UNDER DIFFERENT LOSSES

To further investigate the statistical behavior of reconstruction errors, we perform a cross-over experiment using three representative pixel-level losses: $\ell_1$, $\ell_2$ (MSE), and a hybrid $\ell_1$+SSIM. Fig. 7 visualizes the resulting error distributions via Q–Q plots, where $\ell_1$ corresponds to a Laplace assumption and $\ell_2$ to a Gaussian assumption.

Within the narrow error range $[-0.05, 0.05]$, the empirical errors exhibit moderate alignment with Gaussian or Laplace models. However, in the tails, all cases display pronounced heavy-tailed behavior, with Wasserstein Distance (WD) consistently above $0.7$, indicating systematic mismatches. Under Gaussian scoring (Fig. 7a–c), the $\ell_2$-trained model achieves the best fit ($R^2 = 0.79$, $WD = 0.78$), suggesting that the learned error distribution partially reflects the optimization objective. Interestingly, under Laplace scoring (Fig. 7d–f), the $\ell_2$-trained model still outperforms the $\ell_1$-trained counterpart ($R^2 = 0.91$ vs. $0.77$), revealing that even when the loss is Laplace-motivated, the actual error statistics may deviate substantially from the assumed distribution due to temporal correlation with the single video .

Another noteworthy finding is the discrepancy between fidelity and distributional fit. For example, $\ell_1$+SSIM achieves the highest PSNR (36.52dB), but its distributional alignment under both Gaussian and Laplace assumptions remains inferior. This implies that higher PSNR does not necessarily cor-

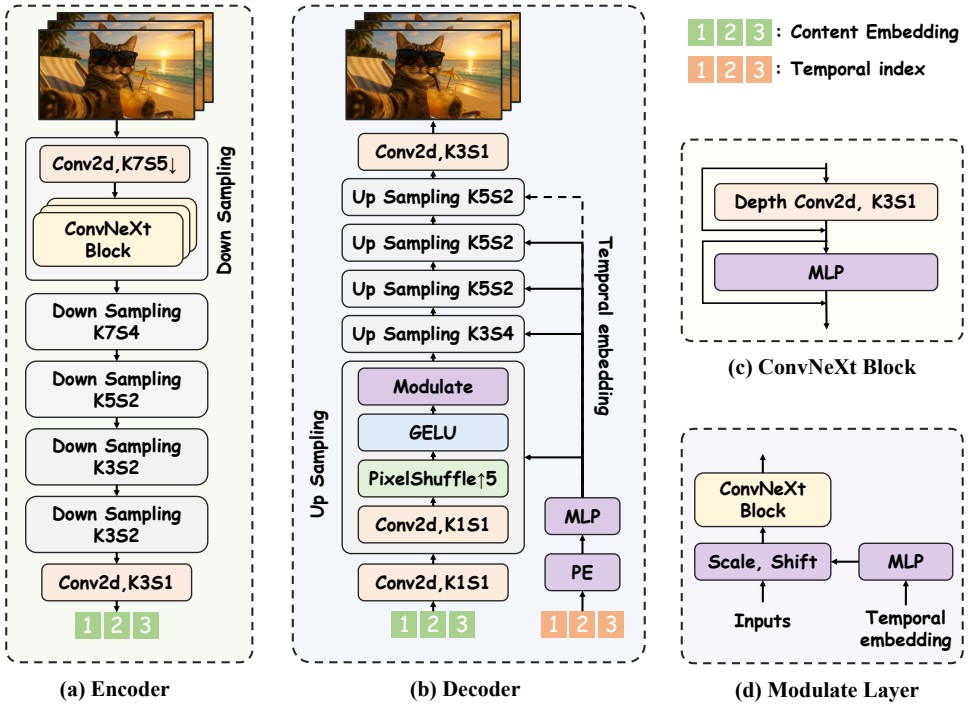

Figure 8: Overview of the proposed architecture. The encoder cascades ConvNeXt (Liu et al., 2022)-based down-sampling blocks to extract compact content embeddings $e_t$ from ground-truth frames. The decoder reconstructs frames from $e_t$ with temporal conditioning: intermediate features are modulated by sinusoidal positional encodings of the time index, enabling adaptive reconstruction across different frames. For efficiency, the last up-sampling layer avoids ConvNeXt operations to handle high-resolution outputs. Strides for different resolutions can be adjusted adaptively.

respond to better statistical consistency. Instead, optimization guided by mismatched error models can lead to sub-optimal convergence behavior.

In summary, pixel-wise losses inherently impose rigid Gaussian or Laplace priors that are systematically violated in single-video INR reconstructions. The empirical error distributions are heavy-tailed, video-dependent, and structurally correlated, rendering simplistic parametric assumptions unreliable. Unlike amortized models such as VAEs, where dataset-level statistics are smoothed over multiple samples, single-video INRs exhibit idiosyncratic error characteristics that challenge conventional pixel-level loss assumptions. These findings motivate exploring alternative optimization domains or perceptually aligned metrics that better capture the intrinsic error structure.

# D ARCHITECTURE

Our model follows an encoder–decoder paradigm with temporal conditioning. The encoder extracts compact content embeddings from input frames, while the decoder reconstructs frames conditioned on both content and temporal indices. The overall architecture is illustrated in Fig. 8.

## D.1 ENCODER

The encoder consists of cascaded down-sampling blocks, each composed of a ConvNeXt block (Liu et al., 2022) followed by a strided convolution for resolution reduction. Given a ground-truth frame $x_t \in \mathbb{R}^{H \times W \times 3}$ at time $t$, the encoder produces a content embedding:

$$e_t = f_{\text{enc}}(x_t) \in \mathbb{R}^{C' \times H' \times W'}, \tag{23}$$

where $H'$ and $W'$ denote the reduced spatial resolution, and $C'$ is the embedding dimension. This formulation is similar to HNeRV (Chen et al., 2023a), but with modified building blocks.

### D.2 TEMPORAL-CONDITIONAL DECODER

The content embedding $e_t$ is fed into a lightweight convolutional decoder to reconstruct the frame:

$$\hat{x}_t = f_{\text{dec}}(e_t) \in \mathbb{R}^{H \times W \times 3}. \tag{24}$$

Following observations from Zhang et al. (2024), we introduce a modulation mechanism to adapt the reconstruction to temporal variations. Given intermediate feature maps $f_t$, temporal modulation is defined as:

$$\text{modulate}(f_t \mid \alpha_t, \beta_t) = \alpha_t(\gamma(t)) \cdot f_t + \beta_t(\gamma(t)), \tag{25}$$

where $\alpha(\cdot)$ and $\beta(\cdot)$ are learned nolinear transformations from a temporal embedding $\gamma(t)$. We adopt sinusoidal positional encodings (Mildenhall et al., 2021; Chen et al., 2021a):

$$\gamma(t) \in \mathbb{R}^{2l} = \left[ \sin(b^0 \pi t), \cos(b^0 \pi t), \dots, \sin(b^{l-1} \pi t), \cos(b^{l-1} \pi t) \right], \tag{26}$$

where $l$ controls the number of frequency components and $b$ is a frequency scaling factor. This encoding mitigates the spectral bias (Xie et al., 2023) of neural networks, allowing the decoder to better capture high-frequency temporal variations.

To reduce computational cost at high resolutions, the final up-sampling layer of the decoder deletes ConvNeXt block, which we found significantly reduces inference latency without degrading quality.

### D.3 RATE IN INR

The rate in INR-based compression is fundamentally determined by two factors: (1) *quality*, referring to the bit-width used to quantize each parameter, and (2) *quantity*, referring to the total number of parameters to be transmitted. Increasing bit-width improves numerical precision and reconstruction fidelity, but directly increases the coding rate. Similarly, enlarging the network size (i.e., parameter count) enhances reconstruction capacity, but also increases the total number of bits to encode. In addition, higher rate inevitably introduces higher computational complexity, since larger models and higher-precision arithmetic both slow down inference and training.

As discussed in Appendix B, this naturally leads to a rate–distortion trade-off. For a fixed architecture, reducing rate (via fewer parameters or lower precision) often sacrifices reconstruction accuracy, while increasing rate improves fidelity at the cost of storage and complexity.

In our implementation, we vary the rate primarily by adjusting the channel dimension of the decoder, while keeping the bit-width fixed, consistent with most prior works (Chen et al., 2023a; Kwan et al., 2023). Although recent studies (Shi et al., 2025) demonstrate that dynamically adjusting bit-width is an effective alternative for rate control, we leave this direction as promising future work.

## E MULTI-VISION MODELS REPRESENTATION

We visualize the feature sensitivity of representative pretrained models on different categories of sequences, including animal, nature, food, building, and face. As shown in Fig. 9, relying on a single pretrained model introduces strong inductive biases toward specific visual patterns. For example, AlexNet (Krizhevsky et al., 2012) consistently emphasizes the main subject region, while VGG (Simonyan & Zisserman, 2014) places greater weight on high-frequency edges and contours. This, to some extent, explains why VGG-based perceptual losses often correlate better with human judgments of visual fidelity, as also observed in Zhang et al. (2018). However, such attention can fail on smooth regions, e.g., VGG tends to under-emphasize facial regions (Fig. 9 (e)) where structural cues are subtle but perceptually critical. In contrast, DINOv2 (Oquab et al., 2023) captures intra-object variability more robustly and yields feature maps that are less biased toward low-level edges or single-object saliency.

These observations highlight that no single pretrained model provides a universally reliable perceptual space. We aggregate feature-based losses from multiple pretrained models. This ensemble

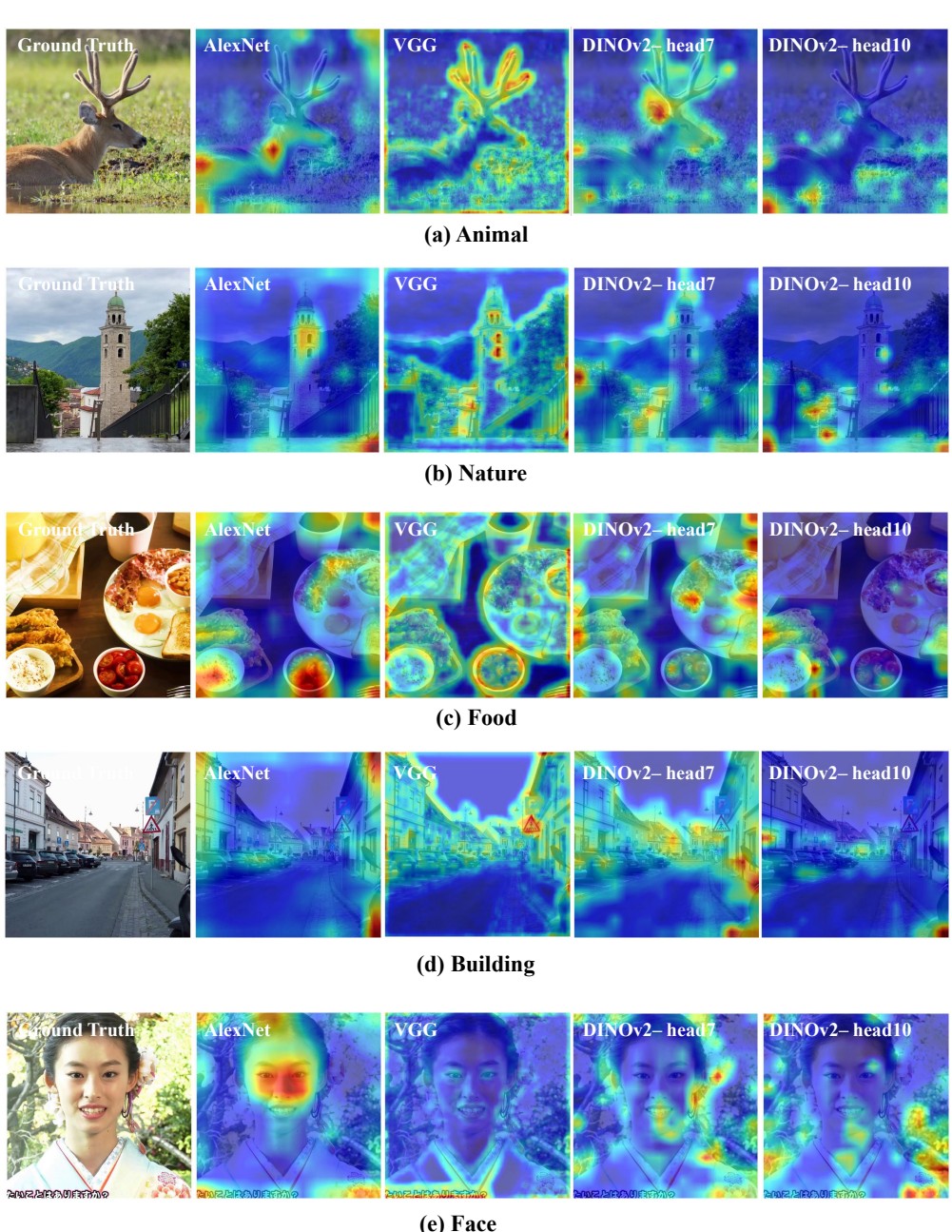

**(a) Animal**

**(b) Nature**

**(c) Food**

**(d) Building**

**(e) Face**

Figure 9: Illustrative heatmaps of feature sensitivity from different pretrained vision models on samples from YouHQ (Zhou et al., 2024). From left to right: AlexNet (Krizhevsky et al., 2012), VGG (Simonyan & Zisserman, 2014), and DINOv2 (Oquab et al., 2023). Each model exhibits distinct inductive biases, emphasizing different spatial or semantic structures, which in turn influences the optimization objective.

strategy reduces the risk of suboptimal alignment that may arise when optimization is guided solely by one model's representational prior.

On a broader perspective, it is worth noting that perceptual transformations do not establish a tractable probabilistic mapping between pixel space and feature space. The pixel-to-feature mapping induced by deep pretrained models is highly nonlinear and analytically intractable, which means that

Gaussian error assumptions in feature space cannot be directly transferred back to pixel space. Only in the case of linear transformations would Gaussianity be preserved across spaces.

# F    EXPERIMENTS

## F.1    IMPLEMENTATION DETAILS

**Test Datasets.**    We evaluate our method on two benchmarks. The UVG dataset[1] comprises seven $1920{\times}1080$ videos at 120 FPS, each 2.5–5 seconds long, serving as a standardized evaluation benchmark. The YouHQ subset (Zhou et al., 2024) contains $\sim$37K high-definition ($1080{\times}1920$) YouTube clips covering diverse content, including street scenes, landscapes, animals, faces, static objects, and nighttime environments. For ablation studies, we center-crop to $960 \times 960$ to reduce spatial size. In short, UVG provides standard evaluation, while YouHQ offers a diverse and lightweight testbed for ablations.

**Training and Implementation.**    Models are trained using Adam (Kingma, 2014) with batch size 1 and an initial learning rate of $1.5 \times 10^{-4}$, scheduled via warmup cosine annealing. Unless stated otherwise, training runs for 150k iterations on UVG and 15k iterations on YouHQ. We apply 7-bit quantization-aware training with straight-through estimator (STE) (Bengio et al., 2013; Yin et al., 2019). All experiments are implemented in PyTorch and executed on NVIDIA RTX A6000 and 4090 GPUs. Code will be released upon publication.

For downsampling, we use Conv2d with width-dependent strides: $(5, 4, 2, 2, 2)$ for inputs of 960 or 1920 pixels, and $(5, 3, 3, 2, 2)$ for 1080 pixels. Upsampling is implemented via PixelShuffle. For experiments involving HNeRV, UVG clips are center-cropped to $1920 \times 960$ to match their setup; otherwise, the original resolution (1080p) is used. Strides for other resolutions can be adjusted adaptively.

**Evaluation Protocol.**    For fairness, we report numbers from published papers when official implementations are unavailable or outdated (e.g., PLVC). For methods with well-maintained open-source code, we reproduce results under our setting for consistency, such as for HNeRV.

## F.2    SEQUENCE-LEVEL EVALUATION.

While conventional frame-level metrics such as PSNR, MS-SSIM, LPIPS, and DISTS provide a direct assessment of spatial fidelity and perceptual quality, they operate independently on individual frames and thus fail to capture temporal dynamics and holistic video realism. To address this limitation, we adopt the sequence-level evaluation protocol from VBench (Huang et al., 2024), which integrates multiple complementary indicators beyond frame fidelity. For clarity, we briefly restate the metrics considered in this work:

1. *Subject Consistency.* Evaluates whether the appearance of a primary subject (e.g., a person, vehicle, or animal) remains stable across frames. This is measured by computing feature similarity with DINO (Caron et al., 2021), which is particularly sensitive to identity variations.

2. *Background Consistency.* Measures temporal stability of backgrounds by comparing CLIP (Radford et al., 2021) embeddings across frames.

3. *Motion Smoothness.* Complements the above appearance-based metrics by quantifying whether object motion follows physically plausible dynamics. This is estimated with motion priors from a video frame interpolation model (Li et al., 2023).

4. *Imaging Quality.* Evaluates perceptual frame-level fidelity by detecting distortions such as over-exposure, noise, or blur. It adopts MUSIQ (Ke et al., 2021), trained on SPAQ (Fang et al., 2020). Although frame-based, it is included here since it forms part of the VBench protocol.

---

[1]Beauty, Bosphorus, HoneyBee, Jockey, ReadySetGo, ShakeNDry, YachtRide

Table 4: Sequence-level evaluation on UVG using VBench (Huang et al., 2024) with 0.01bpp.

| Sequence | Subject Consistency | Background Consistency | Motion Smoothness | Image Quality | Average Score |
|---|---|---|---|---|---|
| Beauty | 96.03% | 94.60% | 99.45% | 56.90% | 86.75% |
| Bosph. | 94.69% | 93.55% | 99.71% | 69.21% | 89.29% |
| Honey. | 99.71% | 98.94% | 99.21% | 63.82% | 90.42% |
| Jockey | 76.76% | 87.65% | 97.74% | 54.86% | 79.25% |
| Ready. | 67.95% | 83.32% | 97.98% | 61.49% | 77.69% |
| Shake. | 74.86% | 88.76% | 99.67% | 65.93% | 82.31% |
| Yacht. | 81.44% | 78.97% | 99.55% | 70.01% | 82.49% |
| Avg. | $84.49_{\pm1.5}\%$ | $89.40_{\pm1.0}\%$ | $99.04_{\pm0.5}\%$ | $63.17_{\pm2.4}\%$ | $84.03_{\pm2.8}\%$ |
| Empirical Min | 14.62% | 26.15% | 70.60% | 0.00% | / |
| Empirical Max | 100% | 100% | 99.75% | 100% | / |

**Note:** We note that sequence-level metrics are sensitive to the number of frames evaluated, since several indicators rely on comparisons relative to the first frame or across temporal neighborhoods. Therefore, consistent evaluation requires using either the full sequence or a standardized subset of frames. In our experiments, we adopt the full set of frames from UVG to ensure reliable and reproducible results.

Here, we provide a detailed sequence-level evaluation in Table 4. The results are consistent with intuitive expectations: sequences with large motion impose significant challenges on maintaining content consistency and motion smoothness. For example, *Jockey* and *ReadySetGo* exhibit substantially lower subject consistency (76.8% and 68.0%, respectively) compared to relatively static sequences such as *HoneyBee* (99.7%) and *Beauty* (96.0%). This corresponds to a reduction of more than 20% in subject stability. A similar trend is observed in background consistency, where motion-heavy sequences (*ReadySetGo* at 83.3%) fall behind stable ones (*HoneyBee* at 98.9%). In contrast, motion smoothness remains consistently high across all sequences (above 97.7%), indicating that INR-based reconstruction preserves local dynamics well, even under large temporal variations.

On the other hand, it is important to recognize that these metrics are all no-reference, and their scores can be influenced by the quality of the original source videos. Nevertheless, they provide valuable insights into temporal behaviors that frame-level metrics overlook. For example, VAE-based methods often exhibit noticeable fluctuations in frame quality, which remain hidden under conventional PSNR or SSIM evaluations. This limitation becomes even more critical under perceptual optimization, where human observers are especially sensitive to temporal inconsistencies. Incorporating sequence-level evaluation thus provides a more holistic and reliable understanding of reconstruction quality in video-INRs.

### F.3 DECODING COMPLEXITY

We evaluate sequential decoding complexity in terms of frames per second (FPS) at 1080p resolution, as shown in Fig. 10. The comparison covers representative approaches: DCVC-RT (Jia et al., 2025), the fastest VAE-based codec with superior efficiency over VVC; DCVC-FM (Li et al., 2024), a state-of-the-art VAE-based codec; VVC and HEVC as conventional baselines; and NVRC (Kwan et al., 2024), a state-of-the-art INR-based codec to date.

Unlike VAE-based codecs, whose decoding FPS remains nearly constant across bitrates due to fixed latent transforms, INR-based methods scale with network evaluation. NVRC achieves strong PSNR gains through rate–distortion optimization and hierarchical grid modeling, but this comes at the cost of significantly lower runtime. In contrast, our method employs a lightweight feed-forward INR optimized with perceptual supervision, yielding far higher FPS while maintaining competitive perceptual quality. For instance, our FP16 implementation with Torch compile reaches over 100 FPS, exceeding the real-time threshold by a large margin, while NVRC remains below 25 FPS.

Most of the decoding cost in our framework stems from the modulation and output layers, especially at high resolutions. Crucially, the absence of inter-frame dependencies enables full decoding paral-

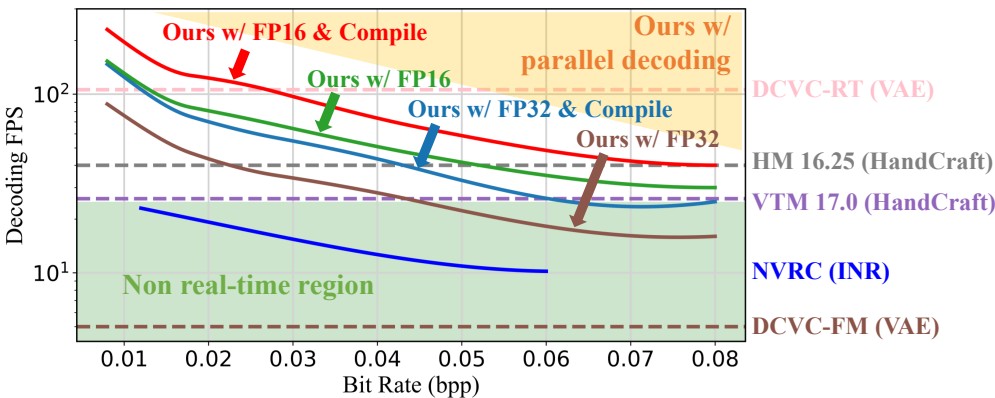

Figure 10: Sequential decoding FPS (frames per second) for 1080p resolution. The reported FPS with *compile* is measured with PyTorch's `torch.compile`, which applies ahead-of-time compilation to optimize model execution by fusing operators and reducing Python overhead. This generally improves runtime performance without altering model accuracy.

lelism: in the ideal case, all frames can be reconstructed within a single forward pass, reducing total latency to that of one model evaluation. As highlighted in Fig. 10, parallel decoding offers an even larger speedup potential, with the achievable FPS depending on the specific implementation (e.g., multi-threading or multi-GPU deployment).

### F.4 VISUALIZATION

To qualitatively evaluate the effect of different optimization strategies, we provide visualization results comparing pixel-domain and feature-domain supervision.

**Pixel Optimization vs. Feature Optimization.** As shown in Fig. 11, under the same architecture, perceptual (feature-domain) optimization yields reconstructions with sharper edges, finer textures, and more faithful perceptual quality. In contrast, pixel-domain supervision tends to produce over-smoothed results that suppress high-frequency details in order to minimize pixel-wise error. This qualitative difference is consistent with our quantitative findings, and highlights the advantage of perceptual optimization in preserving semantically relevant structures.

**Comparison with Other Methods.** We visualize reconstructions on the *Beauty* and *YachtRide* sequences in Fig. 12 and 13. Due to limited availability of some prior works (either not fully open-sourced or difficult to reproduce), we select representative methods for comparison. The images are arranged in order of increasing PSNR for clarity.

As shown, our method achieves the best perceptual quality despite having lower PSNR, highlighting the well-known limitation of pixel-wise metrics in capturing perceptual fidelity. In contrast, DCVC-FM attains the highest PSNR but produces overly smooth textures, illustrating that high numerical fidelity does not necessarily correspond to visually pleasing reconstructions. This comparison underscores the advantage of feature-domain supervision in preserving semantic details and perceptual realism.

However, it is important to note that perceptual optimization may degrade fine-grained details, such as subtle textures or small facial features.

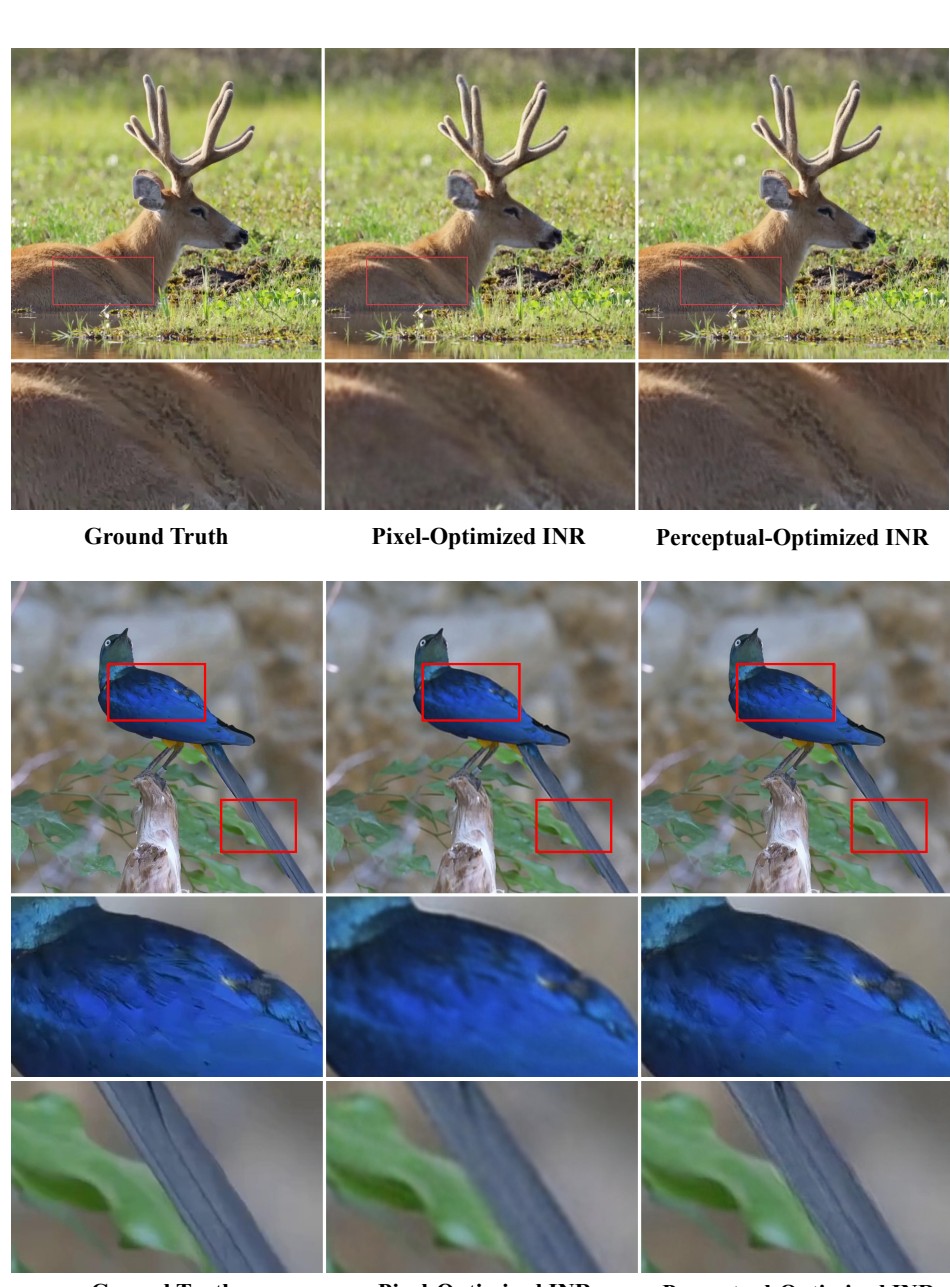

Figure 11: The visualization of our 0.6MB model optimized with different supervision on sample from YouHQ.

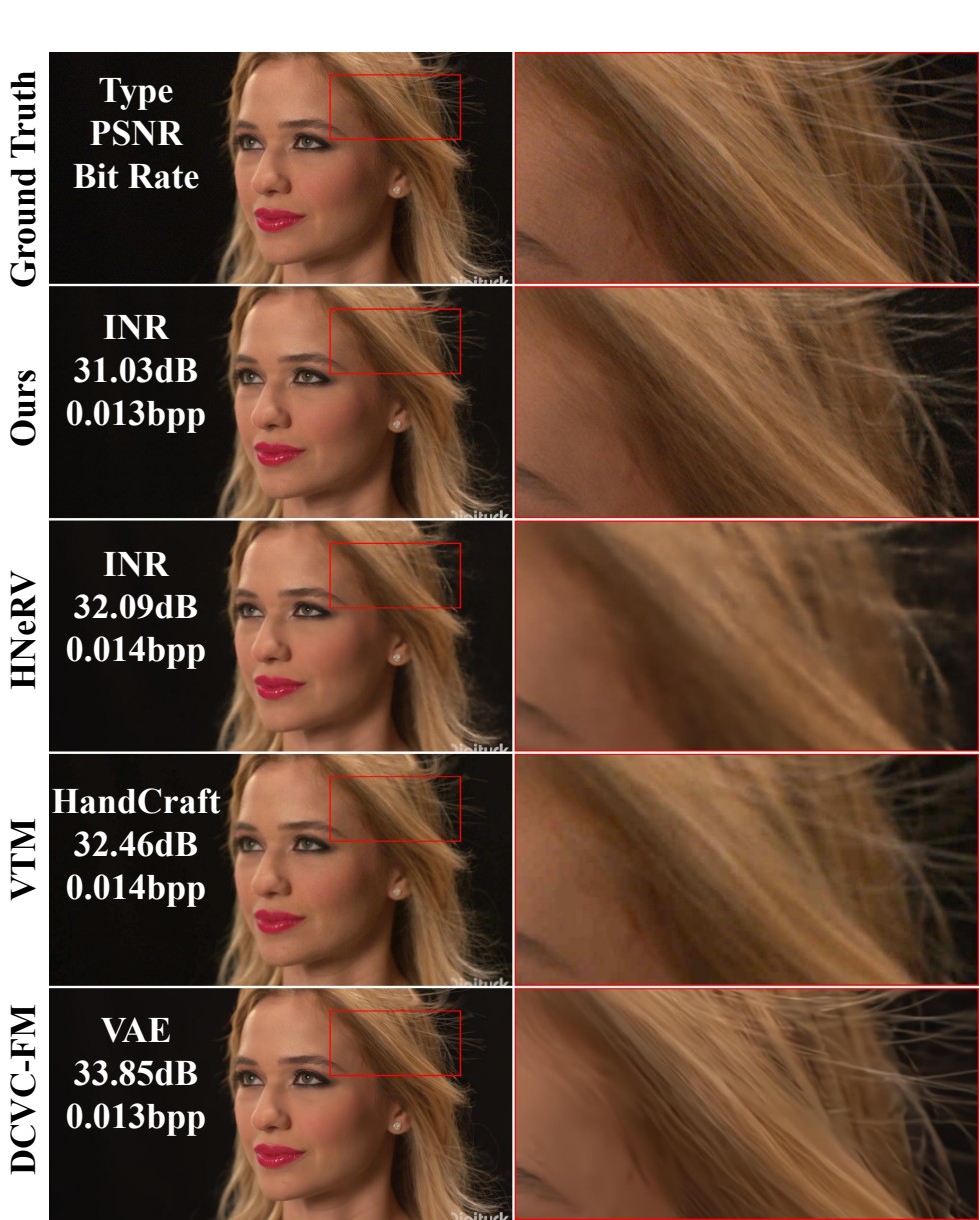

Figure 12: The visualization of *Beauty* sequence.

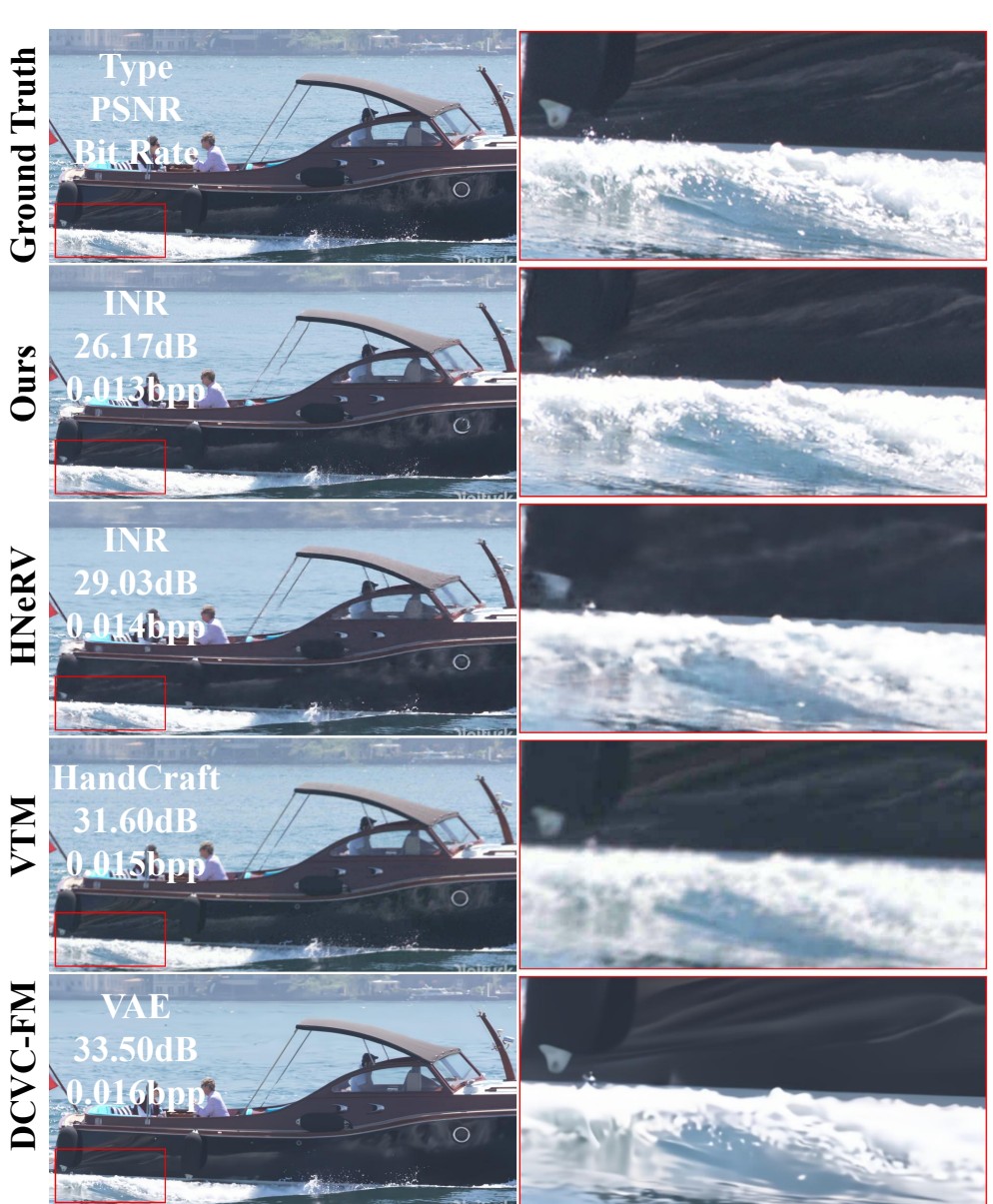

Figure 13: The visualization of *YachtRide* sequence.

