# OpenReview forum: "Rethinking Video-INRs through Perceptual Optimization"
_ICLR.cc/2026/Conference — ICLR 2026 Conference Withdrawn Submission_

### Official Review · Reviewer_3uXh · 2025-10-20

**Soundness:** 3
**Presentation:** 3
**Contribution:** 3
**Rating:** 4
**Confidence:** 4

**Summary:**

## Summary
* This paper propose a perceptual video codec using INR.
* The major contribution of the proposed approach are two loss functions. The first one is based on pre-trained vision model as an improvement of LPIPS. The second one is based on pre-trained self-supervised vision model such as DINO.
* The authors successfully defend their proposed approach by ablation study.

**Strengths:**

## Strength
* The authors propose the first INR based perceptual codec.
* The authors take frame level consistency into consideration, and present benchmark results in Table 2.
* The empirical result is convincing. The authors successfully show the effectiveness of the proposed loss function.

**Weaknesses:**

## Weakness
* One major weakness is that several obvious alternatives to the proposed approach are not considered:
  * First, it appears that LPIPS and DISTS are obvious alternatives to VFS (VGG) in Tab 3. It would be great to include them in comparsion.
  * Second, one obvious alternative to the proposed approach is to train a post processing based conditional generative model on existing NVC methods such as DCVC-FM / NVRC. This approach is already verified in image compression [High-Fidelity Image Compression with Score-based Generative Models] [A Residual Diffusion Model for High Perceptual Quality Codec Augmentation]
* Another major issue is, even if we do not consider the obvious alternatives, the proposed approach does not look complete by itself. The old school way of perceptual image compression, as shown in [High-Fidelity Generative Image Compression] [Improving Statistical Fidelity for Neural Image Compression with Implicit Local Likelihood Models], has two stage. The first stage, similar to this maniscript, is to train a image codec with a mixture of L1 loss and LPIPS. The second stage, is to train a conditional GAN / diffusion based on the encoding result of the first stage. The proposed approach can be seen as an improvement of LPIPS for video model in the first stage. The second stage is missing.
* Minors:
  * The overall objective of the manuscript is aligned with [Good, cheap, and fast: Overfitted image compression with Wasserstein distortion]. It would be great if the authors include it in comparsion.
  * It would be great if the authors can provide a more complete list of metrics in Table 1, including FID, KID and encoding time for a more comprehensive comparsion.
  * It would be great if the authors can include DVC-P, DiffVC in Table 2, to better evaluate the consistency of other perceptual NVC that the authors have compared in Table 1.

**Questions:**

## Questions:
* Is that possible that the proposed approach can be further improved, by training the "second stage" conditional GAN / diffusion based on the current result?
* What are the FID / KID of the proposed approach like? The authors propose somewhat more complex loss functions, does this add a lot to encoding time?

---

### Official Review · Reviewer_hmzM · 2025-10-23

**Soundness:** 3
**Presentation:** 3
**Contribution:** 2
**Rating:** 2
**Confidence:** 4

**Summary:**

The paper proposes a method for training INRs which uses perceptual losses instead of traditional error based losses. The paper first provides a theoretical and empirical discussion that explains the implications of error based losses in terms of assumptions on the distribution of video data which may not hold in practice. The paper then develops a method for single-frame INR with latent code (similar to HNeRV) which is supervised using LPIPS/DISTS along with several neural feature-space losses and some regularization terms. The paper compares with several recent works where it shows good scores on perceptual metrics.

**Strengths:**

- The perceptual metric scores are good
- The decoding time is fast, can achieve real time
- Overall BPP is good
- The simple formulation is applicable to many different techniques

**Weaknesses:**

- The contribution is weak, perceptual objectives for reconstruction are known
- Many hyperparameters: how were they chosen and what were they?
- Missing some comparisons: how does the method compare with other INR methods?
- No discussion of encoding time

**Questions:**

This paper has some interesting analysis which culminates in a fairly known result: that optimizing perceptual metrics for reconstruction purposes yields a better score on perceptual metrics. I do think the analysis has value and I like the results themselves, but the central contribution here does seem quite weak as this is something which has been happening for years in image restoration and autoencoder based learned compression. The final loss function which was used contains five balancing hyperparameters and eight different specific functions (feature spaces, perceptual losses, error losses, etc.) and I didnt note any guidance on how these were chosen, how sensitive the method is to improperly chosen hyperparameters, or how to reproduce the experiments in Section 4, this really needs to be better explained. The final comparisons, while good numerically, were missing some other methods. Specifically other INR methods, I think HNeRV is the only one. It would be good to understand how the method compares to other INRs specifically in terms of decoding FPS since that is a common win for INR methods. And on that note, measuring encoding time is dismissed entirely:

> While video-INR methods are sometimes criticized for high encoding latency that constrains live streaming (Bentaleb et al., 2025), our perceptually optimized INR remains highly suitable for video- on-demand (VOD) (Liu et al., 2024) and large-scale storage applications, where decoding efficiency constitutes the primary bottleneck.

I disagree and I think we need to start looking at encoding time, if at least to report it. Yes, VOD applications are less sensitive to encoding time but that doesn't mean we can spend hours encoding a single video.

Specific questions:
1. How were the hyperparameters chosen and what were they for the experiments reported in the paper?
2. What is the encoding time for this method and on what hardware?
3. How does this method compare against more contemporary INRs in BPP, Image quality, encoding speed, and decoding speed?

---

### Official Review · Reviewer_qHCU · 2025-10-30

**Soundness:** 2
**Presentation:** 3
**Contribution:** 2
**Rating:** 6
**Confidence:** 5

**Summary:**

The paper proposes using perceptual losses to train better INRs for video compression. Concretely, they train with standard pixel loss, SSIM loss, MVFS loss (which is perceptual, LPIPS/DISTS), VSS loss (DINOv2 across frames), and GAN loss. This heavier pipeline increases training cost, and the other losses hurt the final PSNR but help quality as measured by LPIPS and DISTS. The resulting model compares favorably against other compression models trained with perceptual losses in terms of rate and especially decoding speed. The paper also provides a theoretical perspective motivating the use of losses other than simple pixelwise L1 or L2.

**Strengths:**

S1. My evaluation of this paper hinges on the fact that I believe the resulting videos do look better, despite the drop in PSNR. The paper shows that it is possible and useful to neglect per-pixel fidelity in favor of making sure there is frame-level fidelity.

S2. The theoretical motivation seems sound, and provides compelling reasons to address outlier pixels (which are inevitable) with losses other than some pixelwise error.

S3. The paper is easy to read/understand.

**Weaknesses:**

W1. The novelty of the algorithmic contributions is somewhat limited. The combination of several different losses for a reconstruction network might be slightly new for INR, but it isn't new generally. The architecture of the video INR model is essentially an HNeRV encoder with an ENeRV decoder (following ideas from Boosting NeRV).

W2. The method increases training complexity significantly. I need to see proof that given similar training time, it performs noticeably better than other INR methods, considering that at the stated 250 epochs, HNeRV would be significantly under-tuned and part of this will have been due to being trained for much less actual time than this method.

W3. The selection of baselines seems a little strange. NVRC does not have reported perceptual metrics or example frames, which seems like a critical omission. If this is due to lack of code, then HiNeRV should have been used, considering their code is publicly available and very easy to run, and this method significantly outperforms HNeRV. DCVC-RT is mentioned in the paper, but only for complexity. How does the BD-rate compare?

**Questions:**

My initial impression is favorable, but this relies on a charitable assumption that my key question has good answers.

1. How does this method compare to other INR methods that are (1) more recent, (2) have equal training time, and (3) report similar metrics, LPIPS/DISTS or some example frames?

2. How does the method compare to DCVC-RT in terms of size-quality tradeoffs?

3. How do all the different components affect cost? Since simply training an INR longer always increases performance, it's very important to consider this dimension. Things like VSS seem expensive but affect performance only slightly, so I think this comparison is extremely important and needs to be done well, e.g. compare w/ and w/o VSS with equal training time, such that w/o VSS more training iterations/epochs are used.

---

### Official Review · Reviewer_njnu · 2025-11-04

**Soundness:** 3
**Presentation:** 3
**Contribution:** 3
**Rating:** 4
**Confidence:** 5

**Summary:**

The authors introduce an encoder-decoder architecture for video compression/representation with a specific focus on realism. Unlike most reconstruction methods which rely on MSE/L1 for their optimization, here they use perceptual metrics which ensures that even a simple encoder-decoder backbone can surpass SOTA VAE methods.

**Strengths:**

- Identifying that the real problem lies in optimization - and fixing that with appropriate losses which in turn allows us to have smaller architectures and hence faster encode/decode times is a novel contribution.
- The authors justify this with empirical evidence by analyzing gaussian/laplacian error distribution plots.
- The proposed architecture outperforms  prior Video-INR works and existing VAE methods in decode speeds and quality metrics.

**Weaknesses:**

- is the observation that perceptual losses lead to reconstructions with better fidelity restricted to over-fitted neural codecs? Or does it generalize when the system is trained across videos?
- Encoding time: The authors mention encoding time is only useful for live streaming and not for video on demand - which is a fair point. But even then, the encoding complexity matters. For example, we cannot be spending ~30 GPU hours to encode few seconds/minutes. So unless the authors explicitly mention the time it takes to train their method on a single UVG video sequence,  I cannot assess its usefulness.
- Does the encoding time correlate directly with quality and compression? Eg: Does the quantized decoder's quality reduce drastically if it is trained for half the number of iterations?

**Questions:**

- As mentioned above, i would like the authors to provide Encoding times for all datasets/videos used.

---

### Note · Authors · 2025-11-12

I have read and agree with the venue's withdrawal policy on behalf of myself and my co-authors.